# Evaluation of Machine Learning Algorithms for Classification of EEG Signals

Francisco Javier Ramírez-Arias [1,2], Enrique Efren García-Guerrero [1], Esteban Tlelo-Cuautle [3], Juan Miguel Colores-Vargas [2], Eloisa García-Canseco [4], Oscar Roberto López-Bonilla [1], Gilberto Manuel Galindo-Aldana [5] and Everardo Inzunza-González [1,*]

1. Facultad de Ingeniería, Arquitectura y Diseño, Universidad Autónoma de Baja California, Carretera Transpeninsular Ensenada-Tijuana No. 3917, Ensenada 22860, Mexico; francisco.javier.ramirez.arias@uabc.edu.mx (F.J.R.-A.); eegarcia@uabc.edu.mx (E.E.G.-G.); olopez@uabc.edu.mx (O.R.L.-B.)
2. Facultad de Ciencias de la Ingeniería y Tecnología, Universidad Autónoma de Baja California, Blvd. Universitario No. 1000, Valle de las Palmas, Tijuana 21500, Mexico; miguel.colores@uabc.edu.mx
3. Departamento de Electrónica, Instituto Nacional de Astrofísica, Óptica y Electrónica, Luis Enrique Erro No. 1, Santa María Tonanzintla, Puebla 72840, Mexico; etlelo@inaoep.mx
4. Facultad de Ciencias, Universidad Autónoma de Baja California, Carretera Transpeninsular Ensenada-Tijuana No. 3917, Ensenada 22860, Mexico; eloisa.garcia@uabc.edu.mx
5. Facultad de Ingeniería y Negocios, Guadalupe Victoria, Universidad Autónoma de Baja California, Carretera Estatal No. 3, Gutiérrez, Mexicali 21720, Mexico; gilberto.galindo.aldana@uabc.edu.mx
* Correspondence: einzunza@uabc.edu.mx; Tel.: +52-646-152-8244

**Abstract:** In brain–computer interfaces (BCIs), it is crucial to process brain signals to improve the accuracy of the classification of motor movements. Machine learning (ML) algorithms such as artificial neural networks (ANNs), linear discriminant analysis (LDA), decision tree (D.T.), K-nearest neighbor (KNN), naive Bayes (N.B.), and support vector machine (SVM) have made significant progress in classification issues. This paper aims to present a signal processing analysis of electroencephalographic (EEG) signals among different feature extraction techniques to train selected classification algorithms to classify signals related to motor movements. The motor movements considered are related to the left hand, right hand, both fists, feet, and relaxation, making this a multiclass problem. In this study, nine ML algorithms were trained with a dataset created by the feature extraction of EEG signals.The EEG signals of 30 Physionet subjects were used to create a dataset related to movement. We used electrodes C3, C1, CZ, C2, and C4 according to the standard 10-10 placement. Then, we extracted the epochs of the EEG signals and applied tone, amplitude levels, and statistical techniques to obtain the set of features. LabVIEW™2015 version custom applications were used for reading the EEG signals; for channel selection, noise filtering, band selection, and feature extraction operations; and for creating the dataset. MATLAB 2021a was used for training, testing, and evaluating the performance metrics of the ML algorithms. In this study, the model of Medium-ANN achieved the best performance, with an AUC average of 0.9998, Cohen's Kappa coefficient of 0.9552, a Matthews correlation coefficient of 0.9819, and a loss of 0.0147. These findings suggest the applicability of our approach to different scenarios, such as implementing robotic prostheses, where the use of superficial features is an acceptable option when resources are limited, as in embedded systems or edge computing devices.

**Keywords:** EEG; BCI; feature extraction; artificial intelligence; machine learning; deep learning; artificial neural network; mental commands; signal classification; pattern recognition

## 1. Introduction

The central nervous system is composed of the spinal cord and the brain; the human brain resides in the skull and is considered an essential part of the central nervous

system [1]. The human brain is composed of 100 billion neurons on average [2], the joint action of which is responsible for thoughts, actions, and emotional states. The brain is divided into the right and left hemispheres, where the right hemisphere is in charge of regulating the muscular activity of the left side of the human body, while the left hemisphere regulates the activities of the right side [3]. In order to measure and record the brain's activities, different neuroimaging techniques are used, including magnetoencephalography (MEG) [4], electrocorticography (ECoG) [5], intracortical neuronal recording, functional magnetic resonance (fMRI) [6], near-infrared spectroscopy (NIRS) [7], and electroencephalography (EEG), the latter being the neurophysiological technique [8] most widely accepted by the scientific community and the private sector in the development of research in fields such as neuroscience, robotics, home automation, the Internet of Things, education, etc. [9].

Electroencephalography (EEG) is a non-invasive procedure for measuring the electrical activity generated within the brain as a result of different mental processes [10]. The electrical signals are acquired through electrodes placed on the scalp's surface; thus, waves with different amplitudes and frequencies that refer to a person's mental state are obtained [11]. The frequency ranges span from 0 Hz to 100 Hz. Based on these ranges, the signals are classified as follows: delta, which ranges from 0 Hz to 4 Hz; theta, which contains signals from 4 Hz to 7 Hz; alpha, where the information range is between 8 Hz and 12 Hz; beta, where the range is between 12 Hz and 30 Hz; and gamma, with a range that covers from 30 Hz to 100 Hz [12,13]. Different ranges of signals are essential for identifying different clinical problems, such as schizophrenia [14], Alzheimer's, insomnia, epileptic disorders, brain tumors, and different injuries and infections related to the central nervous system. Furthermore, classification of motor impairment in neural disorders by means of EEG signals processing has been a successful method for identifying central nervous system roots of motor disabilities [15]. Compared with other methods, this neuroimaging technique offers advantages such as portability, temporal resolution, safety, cost, small time constants, simple equipment, and effectiveness [16].

The EEG neuroimaging method is the preferred method for developing brain–computer interfaces (BCIs), both in the academic community and the private sector. Historically, BCI has been clinically applied for understanding motor impairment, both in verbal communication [17] and limb movement [18], as well as cognitive impairment [19], and offers a great advantage over electromyography pattern recognition [20] due to the lack of neuromuscular signals under amputation conditions. BCIs are direct communication and control channels between users' brains and computers where muscle activity is not involved [21,22]. They are currently considered a powerful communication technology as they do not involve muscular routes to complete tasks such as communication, commands, and actions. The basis of these systems is the computer, whose central role is the analysis of EEG signals [23,24]. BCIs are classified as exogenous and endogenous. Exogenous BCIs require external conditions or stimuli so that the brain can generate a particular response based on the stimulus. Endogenous BCIs do not require external stimulation; however, they require some training on the user's part so that they can regulate brain rhythms [24]. Despite the differences mentioned, most BCI models contain the following elements: signal acquisition, information preprocessing, feature extraction, and classification [16,25]. The acquisition of signals is carried out by employing electrodes placed on the scalp's surface [26], through which analog signals are obtained and then digitized by means of analog–digital converters. The next step is the preprocessing of the signals, whereby the following are removed: noise induced by the electrical line; the background noise of the brain; various artifacts that the EEG signals present as a result of some muscular activity such as eye movement, facial muscle activity, etc. [27]. Feature extraction is one of the crucial steps due to its impact on the performance of classification algorithms [28]. Some of the obtained features are in the domains of time and frequency [28], i.e., mean, median, variance, maximum, and minimum, among others [29,30]. The feature extraction process produces a vector containing the most relevant features of the EEG signals, used as input

for classification algorithms. The next step is classification, which is carried out by different algorithms, including LDA, SVM [31], KNN [32], D.T. [33], N.B., and ANN [34].

Currently, there are different fields of science, engineering, and research that evaluate and make use of BCIs to develop applications that present solutions to complex problems [35,36]. These have been possible due to advances in high-density electronics, data acquisition systems that allow high-quality EEG signals to be acquired, intelligent systems that use machine and deep learning algorithms, and neural networks that allow pattern recognition and signal classification to be performed with high precision. In [25], the authors explain that BCIs can be used in the following six application scenarios: replace, restore, augment, enhance, supplement, and research tools. The authors of [37] commented that current and future BCI application areas are device control, user status monitoring, assessment, training and education, gaming and entertainment, cognitive enhancement, safety, and security. Intelligent systems commonly incorporate machine learning (ML) approaches [38–40]. ML refers to a system able to learn from training data from certain activities so that the analytical model generation process is automated, and associated tasks can be completed or supplemented [41,42]. Deep learning (DL) is a paradigm within ML based on the use of artificial neural networks (ANNs) [41]. Commonly, ML algorithms focus on classifying EEG signals related to the motor and imaginary movements of hands and feet to carry out control actions, as presented in [43–46]. DL is useful in areas with vast and high-dimensional data; therefore, deep neural networks outperform ML algorithms for most text, images, video, voice, and audio processing techniques [47]. Nevertheless, for low-dimensional data input, especially with insufficient training data, ML algorithms may still achieve superior results [48], which are even more interpretable than deep neural network results [49]. The authors of [50] used power, mean, and energy as features to classify EEG signals related to the right and left hands through artificial neural networks (ANNs) and support vector machine (SVM). In [51], the authors used SVM to control the direction of a wheelchair by extracting the mean, energy, maximum value, minimum value, and dominant frequency characteristics of the EEG signals. In [52], the authors used the fast Fourier transform and principal component analysis as characteristics of the EEG signals to feed the SVM classifier to control a robotic arm. The authors of [53] reported the use of EEG signals to control an exoskeleton and the use of SVM, LDA, and NN for their respective classification. Studies such as the one presented in [54] have used pretrained neural network models to classify EEG signals through time–frequency characteristics. Recent studies have focused on the proper selection of EEG signal characteristics and its effect on the accuracy of ML and DL algorithms, as presented in [30]. ML and DL techniques are widely accepted and help to develop specific tasks within different applications [55–61]. Moreover, they are increasingly used to obtain EEG data for pattern analysis, classification of group membership, and BCIs [29,62–67]. However, there are still open research problems, such as the real-time processing of EEG signal classification and the optimization of ML algorithms for implementation on embedded systems or edge computing devices. Hence, research on and development of reliable, efficient, and robust systems for EEG signal classification, among others, should be pursued [16,68]. The complexity of human movements for the manipulation of tools is very high and diverse; for an adult human brain that has automated different movements, it does not represent a major effort, however, for ML it requires the management of precise information inputs that allow programming and execution of free movement. Previous studies offer multiple classes of motor imagery limb movements based on EEG spectral and time domain descriptors [69]; in this sense, there continues to be a need in machine learning to increase the reliability and accuracy of EEG signals used for programming human-like movements.

For the reasons stated above, the aim of this paper is to evaluate nine ML algorithms for the classification of EEG signals. The purpose is to find which ML model presents the best performance metrics for the identification of movement patterns in EEG signals for the control of a mechatronic system, in this case, a robotic hand prosthesis. The selected dataset consists of more than 1500 EEG recordings of 1–2 min in length from 109 subjects

and is publicly available in [70]. In this study, we randomly selected 30 subjects to train, validate, and test the proposed method. The ultimate aim is to facilitate the development of robotic limb prosthetics, which is possible because ML algorithms can recognize patterns in EEG signals with complex dynamics. The hypothesis is that ML algorithms perform better in tasks of signal classification than standard methods. The novelty of this study is to provide a methodology for the classification of EEG signals by training several ML algorithms and employing processing, analysis, and feature extraction techniques in the time domain of various lapses of EEG signals related to motor tasks, which can be translated into commands for the control of mechanisms or mechatronic systems such as wheelchairs, robotic prostheses, and mobile robots.

The rest of this paper is organized as follows: Section 2 presents the materials used for the development of the proposed method; additionally, the description of the dataset used for this paper is presented. Section 3 presents the performance metrics obtained from the proposed ML models and the discussion of the main findings obtained in this study. Section 5 presents the proposed usage scenario in a real-world application. Conclusions and future work are described in Section 6.

## 2. Materials and Methods

### 2.1. Hardware and Software

The hardware used for the implementation of the proposed method had the following specifications: Microsoft Windows 10 Pro operating system, system model OptiPlex 3070, system type ×64-based PC, Processor Intel Core i5-9500 at 3.00 GHz, six Cores, six logical processors, memory (RAM) of 16.0 GB DDR4 2666 MHz (2 × 8 GB), and NVIDIA GeForce GT 1030 GDDR5 2 GB PCI-Express ×16. The software used for reading the EEG signals, electrode selection, signal segmentation, preprocessing, analysis, feature extraction, and preparation of the dataset was LabVIEW 2015. Furthermore, the following libraries, which are part of the development environment of LabVIEW, were used: Biomedical Toolkit and Signal Express. The MATLAB 2021a version was used for training and testing the different ML algorithms, which are part of the Statistics, Machine Learning and Deep Learning Toolbox.

### 2.2. Machine Learning Algorithm Training

In this paper, we selected nine ML algorithms to evaluate their performance in the classification of EEG signals related to the motor movements of right hand, left hand, both fists, feet, and relaxation. The nine selected algorithms are naive Bayes (N.B.), k-nearest neighbors (KNN), decision tree (D.T.), support vector machine (SVM), linear discriminant analysis (LDA), Narrow-ANN, Medium-ANN, Wide-ANN, and Bilayered-ANN. These ML algorithms are part of the statistical and machine learning toolbox of MATLAB, which has various tools that can be used for both the pre- and post-processing of data.

Figure 1 shows the block diagram to train, test, and evaluate the selected ML algorithms. First, the dataset is loaded; the chosen dataset is constituted of more than 1500 EEG recordings from 109 subjects that become between 1 and 2 minutes long and can be found in [70]. In this study, 30 people were randomly chosen to train, test, and validate the proposed method. Subsequently, the data are normalized between 0 and 1 to obtain better results. Next, we randomly split the dataset into 80% for training and 20% for testing. Then, the ML model is trained. The next step is to obtain the performance metrics of the ML models (for example, using the confusion matrix), i.e., the performance metrics to evaluate the ML algorithms, such as the area under the curve (AUC) and accuracy, among others.

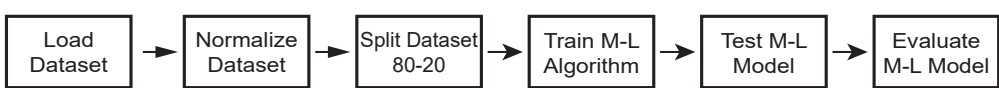

**Figure 1.** Block diagram for training, testing, and evaluating the ML algorithms.

A typical system for EEG signal classification is conceptually divided into signal acquisition, preprocessing, feature extraction, and classification [23,71]. The EEG signals are acquired by electrodes located on the scalp's surface that transfer information on the electrical neuronal activity to the data acquisition system. In preprocessing, line noise and muscle artifacts are removed from EEG signals. Feature extraction uses several digital signal processing techniques to obtain feature vectors. These vectors are used to train the ML or DL algorithms to classify the EEG signals. The result of the algorithms is a specific class, as illustrated in Figure 2. The following subsections describe the procedure in detail.

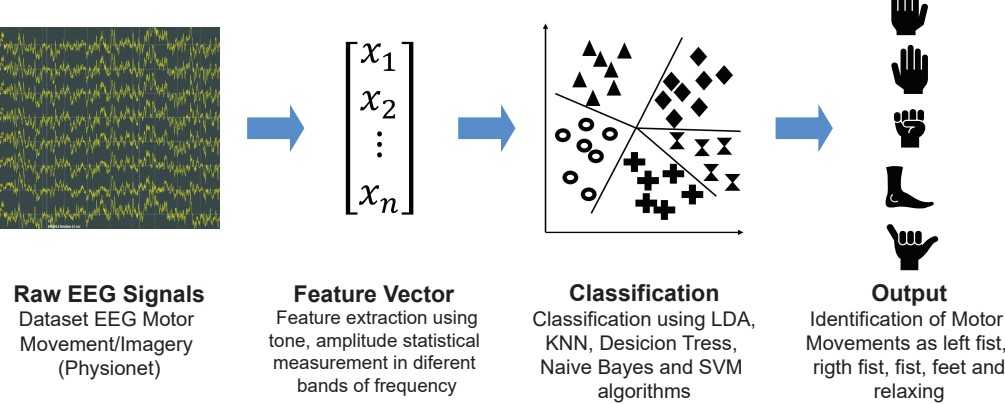

**Raw EEG Signals**
Dataset EEG Motor Movement/Imagery (Physionet)

**Feature Vector**
Feature extraction using tone, amplitude statistical measurement in diferent bands of frequency

**Classification**
Classification using LDA, KNN, Desicion Tress, Naive Bayes and SVM algorithms

**Output**
Identification of Motor Movements as left fist, rigth fist, fist, feet and relaxing

**Figure 2.** Proposed method for classifying EEG signals.

### 2.3. Input Data

The dataset used for EEG signal classification was developed by Schalk and colleagues at Nervous System Disorders Laboratory and is publicly available on Physionet [70]. The data consist of more than 1500 EEG recordings of 1–2 min in length from 109 subjects. Patients performed 14 tasks (experiments) while 64 electrodes acquired and recorded the EEG signals through the BCI2000 system [72]. The data are in EDF+ format [73], and they contain 64 EEG signals, each displayed at a rate of 160 samples per second, and an annotation channel, which refers to the actions performed during the task. Table 1 shows the protocol of the Schalk agreement experiment. The diagram of the position of the electrodes used to record the data is the standard 10-10 placement. The dataset consists of 109 folders, and each folder contains 28 files, where 14 of these have the *.edf* extension, and the other 14 have the *.edf.event* extension. The files that contain the EEG signals are those that contain the *.edf* extension. The *.edf.event* files refer to the events during the development of the different tasks. Although the original set of recorded data consists of continuous multichannel data, and the number of users that comprise it is extensive, we only used the EEG signals of 30 randomly selected subjects, and the tasks that are related to the real movements that take place in tasks 3, 5, 7, 9, 11, and 13. In tasks 3, 7, and 11, real movements related to the right and left fists and relaxation are carried out, while in tasks 5, 9, and 13, real movements of both fists and both feet are carried out. Table 1 summarizes the dataset used in the proposed approach.

**Table 1.** Tasks presented in the dataset to train the ML algorithms for EEG signal classification.

| Task | Real Movement | Imaginary Movement | To | T1 | T2 | Duration |
|------|---------------|---------------------|----|----|----|----------|
| 1 | Open Eyes | - | Relaxing | - | - | 1 min |
| 2 | Close Eyes | - | Relaxing | - | - | 1 min |
| 3 | Fist | - | Relaxing | Left | Right | 2 min |
| 4 | - | Fist | Relaxing | Left | Right | 2 min |
| 5 | Fist/Feet | - | Relaxing | Fist | Feet | 2 min |
| 6 | - | Fist/Feet | Relaxing | Fist | Feet | 2 min |
| 7 | Fist | - | Relaxing | Left | Right | 2 min |
| 8 | - | Fist | Relaxing | Left | Right | 2 min |
| 9 | Fist/Feet | - | Relaxing | Fist | Feet | 2 min |
| 10 | - | Fist/Feet | Relaxing | Fist | Feet | 2 min |
| 11 | Fist | - | Relaxing | Left | Right | 2 min |
| 12 | - | Fist | Relaxing | Left | Right | 2 min |
| 13 | Fist/Feet | - | Relaxing | Fist | Feet | 2 min |
| 14 | - | Fist/Feet | Relaxing | Fist | Feet | 2 min |

*2.4. Proposed Method for EEG Signal Processing*

Figure 3 depicts the proposed method for EEG signal processing, described in detail in the following subsections.

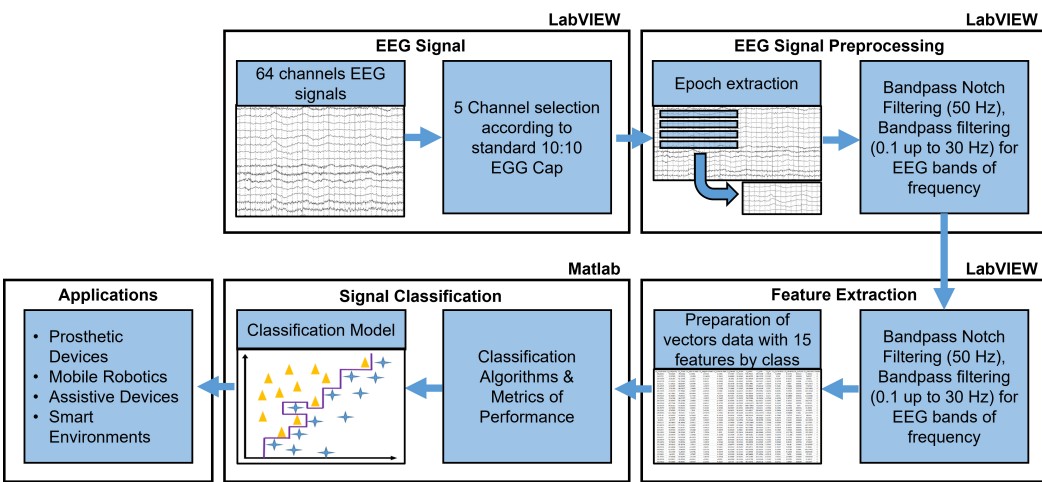

**Figure 3.** Proposed method for EEG signal classification.

*2.5. EEG Signal Acquisition and Channel Selection*

The LabVIEW software 2015 version was employed as the development platform, while the Biomedical Toolkit was used to import the EEG signals, due to the signals being in EDF format. The selected electrodes are shown in Figure 4b. These electrodes present neuronal activity correlated to the execution of the left- and right-hand movements (contained in electrodes C3, C4, and CZ [74,75]) and the neuronal activity related to the movement of both feet (contained in electrodes C1 and C2 [76]); because the different EEG channels tend to represent redundant information, as mentioned in [77], electrodes C3, C1, CZ, C2, and C4 were selected in our study. The selected electrodes were located around the center of the skull, within the motor cortex area; their characteristic is that these electrodes are the least affected by different artifacts [78], which allows the reliable extraction of features to be obtained.

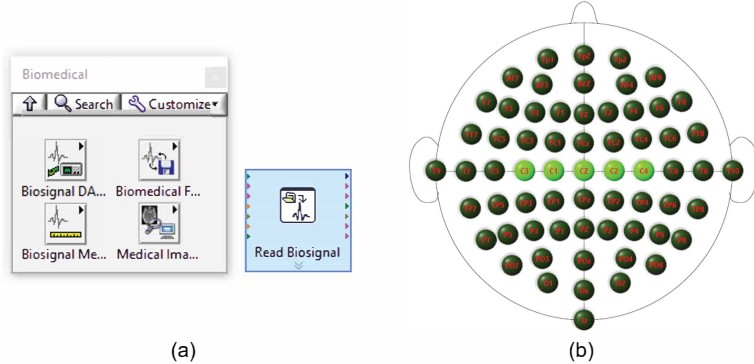

(a)

(b)

**Figure 4.** Selected electrodes for EEG signal classification. (**a**) Biomedical toolkit and (**b**) electrodes selected.

## 2.6. Preprocessing

The EEG signals used, with a sampled frequency of 160 Hz, are available online [70]. Bandpass filters were required to select only the frequencies of interest and eliminate line noise and some other interferences. For this study, we processed the EEG signals through an IIR bandpass filter, with third-order Butterworth topology from 0.1 to 50 Hz. After this, a 50 Hz notch filter was applied to the signals to eliminate noise from the signal power line. Figure 5 shows the original readings of the electrodes used before and after applying the different filters related to the signal preprocessing operations.

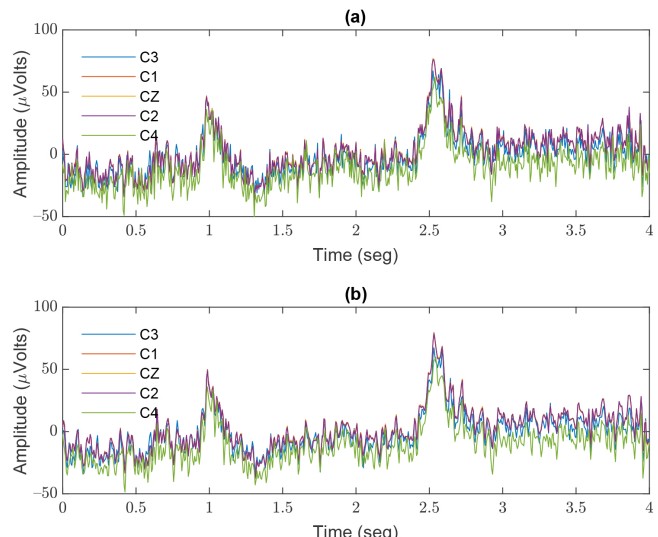

**Figure 5.** EEG signals acquired from electrodes C3, C1, CZ, C2, and C4. (**a**) Original EEG signal and (**b**) filtered EEG signal.

## 2.7. EEG Band Separation

Within EEG signal analysis, it is common to separate a signal into different frequency bands, including Delta (1–4 Hz), Theta (4–8 Hz), Alpha (8–12 Hz), Beta (12–30 Hz), and Gamma (30–50 Hz). As shown in Table 2, third-order bandpass Butterworth IIR filters with different cut-off frequencies were used to carry out this separation.

**Table 2.** Cut-off frequencies of bandpass filters for band extraction of EEG signals.

| Band of EEG Signal | Low Cut-Off Frequency | High Cut-Off Frequency |
|---|---|---|
| Delta | 0.1 Hz | 3.99 Hz |
| Theta | 4.0 Hz | 7.99 Hz |
| Alpha | 8.0 Hz | 11.99 Hz |
| Beta | 12.0 Hz | 29.99 Hz |
| Gamma | 30.0 Hz | 49.99 Hz |

*2.8. Feature Extraction*

The features of the EEG rhythm can be obtained by using several digital signal processing techniques. These features were used for training the nine ML algorithms. These analysis techniques included measurements of tone, amplitude, and level, as well as statistical analyses. Table 3 shows the type of measurements and features obtained when these techniques were applied to the EGG signal epochs.

**Table 3.** Features of the EEG signal used to train the ML algorithms.

| | | | | | | | | | | | | | | | | |
|---|---|---|---|---|---|---|---|---|---|---|---|---|---|---|---|---|
| **Features of the Channels for the Different Electrode Positions** | | | | | | | | | | | | | | | | |
| **Band** | **C3** | **C3** | **C3** | **C1** | **C1** | **C1** | **Cz** | **Cz** | **Cz** | **C2** | **C2** | **C2** | **C4** | **C4** | **C4** | **Class** |
| Delta | Amplitude | Frequency | Phase | Peak to Peak | Neg.Peak | Pos.Peak | Median | Mode | Mean | RMS | S.D. | Summation | Variance | Kurtosis | Skewness | Relaxing |
| Theta | Amplitude | Frequency | Phase | Peak to Peak | Neg.Peak | Pos.Peak | Median | Mode | Mean | RMS | S.D. | Summation | Variance | Kurtosis | Skewness | Left Hand |
| Alpha | Amplitude | Frequency | Phase | Peak to Peak | Neg.Peak | Pos.Peak | Median | Mode | Mean | RMS | S.D. | Summation | Variance | Kurtosis | Skewness | Right Hand |
| Beta | Amplitude | Frequency | Phase | Peak to Peak | Neg.Peak | Pos.Peak | Median | Mode | Mean | RMS | S.D. | Summation | Variance | Kurtosis | Skewness | Fist |
| Gamma | Amplitude | Frequency | Phase | Peak to Peak | Neg.Peak | Pos.Peak | Median | Mode | Mean | RMS | S.D. | Summation | Variance | Kurtosis | Skewness | Feet |

Signal Analysis
- Tone measurements. The tone measurements carried out in the EEG signal epochs were the following: amplitude, frequency, and phase.
- Level measurements. The level measurements implemented in the EEG signal epochs were the following: peak-to-peak, negative peak, and positive peak.
- Statistical features. The statistical measurements applied to the different signal epochs were the following:
  - Median [30,79]:

$$Median = \begin{cases} \frac{(N+1)}{2}, & \text{when } N \text{ is odd} \\ \frac{N}{2} + \frac{(N+1)}{2}, & \text{when } N \text{ is ever} \end{cases} \qquad (1)$$

- Mode is the number that occurs most frequently in the set;
- Mean [80]:

$$\widetilde{x} = \frac{1}{N} \sum_{i=1}^{N} x_i \tag{2}$$

- Root mean square (RMS) [80]:

$$RMS = \sqrt{\frac{1}{N} \sum_{i=1}^{N} x_i^2} \tag{3}$$

- Standard deviation [80]:

$$S = \sqrt{\frac{1}{N} \sum_{i=1}^{N} (x_i - \widetilde{x})^2} \tag{4}$$

- Summation:

$$\sum_{i=1}^{N} x_i \tag{5}$$

- Variance [80]:

$$S^2 = \frac{1}{N} \sum_{i=1}^{N} (x_i - \widetilde{x})^2 \tag{6}$$

where $\widetilde{x}$ is the mean;
- Kurtosis [80]:

$$Kurtosis = \sum_{i=1}^{N} \frac{(x_i - \widetilde{x})^4}{(N-1)s^4} \tag{7}$$

- Skewness [80]:

$$Skewness = \sum_{i=1}^{N} \frac{(x_i - \widetilde{x})^3}{(N-1)s^3} \tag{8}$$

### 2.9. Dataset Preparation

The data vectors consist of 15 features, 3 features for each electrode; the electrodes correspond to positions C3, C1, CZ, C2, and C4, which are related to motor movements, and these belong to one of the five classes of "relaxation", "Right hand", "Left hand", and "Fist and Feet". The dataset has 2792 samples, where 558 samples correspond to the "Relaxation" class, 567 to the right hand, 555 to the left hand, 561 to both fists, and 547 to the feet. On average, there are 557 samples per class, which preserves the balance among the classes. Figure A1 in Appendix A shows a fragment of the dataset created by processing EEG signals when different users performed different motor tasks. Figure A2 in Appendix B depicts the graphic user interface (GUI) of the software (App) developed for the feature extraction process. The proposed App allows features to be extracted in different frequency bands, where each frequency band corresponds to a different class. Table 3 shows the features obtained for training the different ML algorithms. Each line represents a vector of features consisting of five electrodes. Three different measurements were made for each electrode, which resulted in a vector with 15 different characteristics used for the training and testing of the ML and DL models. It can be observed that the feature vector is labeled with its respective class. For each of the five classes, 15 different features were obtained in five different frequency bands to improve the classification accuracy of the ML algorithms [81,82]. The proposed dataset can be downloaded at the link from Supplementary Materials.

## 3. Results

To evaluate the performance of the ML algorithms, we used the following scoring metrics: *Accuracy*, *Error*, *Recall*, *Specificity*, *Precision*, and *F1-Score*. The performance evaluation of the proposed ML models was initiated by calculating *Sensitivity*, *Specificity*, *Precision*, and *Accuracy* [83,84]. *Sensitivity*, also known as *Recall* [84], measures the proportion of positives that are correctly identified as such; it can be calculated by (9). Similarly, *Specificity* measures the proportion of negatives that are correctly identified as such [84]; it can be calculated by (10). *Precision* is the proportion of true positives among the positive predictions [84]; it can be calculated by (11). *Accuracy* can be calculated using (12):

$$Recall = \frac{TruePositives}{FalseNegative + TruePositives}, \tag{9}$$

$$Specificity = \frac{TrueNegatives}{FalsePositives + TrueNegatives}, \tag{10}$$

$$Precision = \frac{TruePositives}{TruePositives + FalsePositives}, \tag{11}$$

$$Accuracy = \frac{TruePositives + TrueNegatives}{TruePositives + FalsePositives + TrueNegatives + FalseNegatives}. \tag{12}$$

*F1-Score* is a method for combining *Precision* and *Recall* into a single measure that includes both [85]. Neither *Accuracy* nor *Recall* can analyze the complete situation on their own. We might have outstanding *Precision* but poor *Recall*, or vice versa, poor *Precision* but good *Recall*. With *F1-Score*, one can represent both concerns with a single score [86]. Once *Accuracy* and *Recall* for a binary or multiclass classification task have been computed, the two scores may be combined to calculate the *F1-Score* metric; it can be calculated by (13):

$$F1 - Score = \frac{2 * Precision * Recall}{Precision + Recall}. \tag{13}$$

Equations (9)–(13) are valid for binary classification and multiclass issues; however, when used for multiclass problems, they must be calculated for each class and then averaged to obtain each metric per model.

Table 4 shows the average scores obtained in each performance metrics by the nine ML algorithms selected in this study. The first parameter analyzed was accuracy, where the LDA model presented an accuracy score of 0.9229; D.T. obtained 0.9803; KNN obtained 0.8996; N.B. obtained 0.9373; SVM obtained 0.9803; Narrow-ANN, Medium-ANN, and Bilayered-ANN obtained 0.9857; finally, Wide-ANN obtained 0.9821. The Narrow-ANN, Medium-ANN, and Bilayered-ANN models obtained the best accuracy score (0.9857). Regarding the error metric, we can see that the LDA, D.T., N.B., SVM, Narrow-ANN, Medium-ANN, Wide-ANN, and Bilayered-ANN algorithms achieved a score less than 0.1, while the KNN model obtained an Error greater than 0.1; therefore, the models with the lowest error were Narrow-ANN, Medium-ANN, and Bilayered-ANN (0.0143). Considering the recall parameter, we observed that the Narrow-ANN algorithm presented the highest score of 0.9863, while the KNN algorithm obtained the lowest score of 0.9037. Regarding the specificity metric, all the algorithms achieved a score greater than 0.9; the ML models with the best results were the Narrow-ANN, Medium-ANN, and Bilayered-ANN models, all scoring 0.9964. Regarding the precision metric, the Bilayered-ANN algorithm is the one that presented the best result, with 0.9859, while the KNN algorithm presented the lowest score, with 0.9099. Regarding the F1-score parameter, the LDA, D.T., N.B., SVM, Narrow-ANN, Medium-ANN, Wide-ANN, and Bilayered-ANN algorithms achieved scores greater than 0.91, while the KNN model obtained a score below 0.91. The algorithm that presented the best F1-score result was Narrow-ANN, with 0.9859.

**Table 4.** The average score parameters of the EEG classification algorithms.

| | Average Scoring Parameters | | | | | |
|---|---|---|---|---|---|---|
| **ML Algorithm** | **Accuracy** | **Error** | **Recall** | **Specificity** | **Precision** | **F1-Score** |
| LDA | 0.9229 | 0.0771 | 0.9219 | 0.9807 | 0.9332 | 0.9228 |
| D.T. | 0.9803 | 0.0197 | 0.9777 | 0.9951 | 0.9792 | 0.9783 |
| KNN | 0.8996 | 0.1004 | 0.9037 | 0.9747 | 0.9099 | 0.9047 |
| N.B. | 0.9373 | 0.0627 | 0.9384 | 0.9844 | 0.9382 | 0.9378 |
| SVM | 0.9803 | 0.0197 | 0.9789 | 0.9950 | 0.9827 | 0.9803 |
| Narrow-ANN | 0.9857 | 0.0143 | 0.9863 | 0.9964 | 0.9857 | 0.9859 |
| Medium-ANN | 0.9857 | 0.0143 | 0.9854 | 0.9964 | 0.9856 | 0.9855 |
| Wide-ANN | 0.9821 | 0.0179 | 0.9834 | 0.9955 | 0.9824 | 0.9828 |
| Bilayered-ANN | 0.9857 | 0.0143 | 0.9854 | 0.9964 | 0.9859 | 0.9856 |

Table 5 presents the performance metrics achieved by each ML algorithm. The metrics used to evaluate the performance of the ML algorithms were the area under the average curve (AUC average), Cohen's Kappa coefficient [87], Matthews correlation coefficient [88], and model loss. Concerning the AUC average metric, all algorithms achieved a score greater than 0.90, where the top three ML models were the SVM, Medium-ANN, and Bilayered-ANN models, which obtained the highest scores (AUC scores). Regarding Cohen's Kappa coefficient, a score above 0.8 indicates exemplary commitment, while zero or less indicates poor commitment. The LDA and KNN algorithms obtained Cohen's Kappa coefficients less than 0.80 but greater than zero, while the D.T., N.B., SVM, Narrow-ANN, Medium-ANN, Wide-ANN, and Bilayered-ANN algorithms achieved Cohen's Kappa coefficients of 0.9384, 0.8040, 0.9384, 0.9552, 0.9552, 0.9440, and 0.9552, respectively, where the Narrow-ANN, Medium-ANN, and Bilayered-ANN algorithms achieved the highest scores. In addition, we used the Matthews correlation coefficient, which has been widely used as a performance metric for ML algorithms since 2000. The best scores obtained were presented by the D.T, N.B., SVM, Narrow-ANN, Medium-ANN, Wide-ANN, and Bilayered-ANN models (0.9736, 0.9225, 0.9757, 0.9824, 0.9819, 0.9783, and 0.9820, respectively), with Narrow-ANN obtaining the best score, while the KNN algorithm achieved the lowest score of 0.8810. The ML model with the lowest loss was Narrow-ANN, with 0.0136, followed by the Medium-ANN and Bilayered-ANN models, both with 0.0147, while the ML algorithm with the highest loss was KNN.

**Table 5.** Performance metrics of the nine ML algorithms trained for EEG signal classification.

| | Performance Metrics | | | |
|---|---|---|---|---|
| **ML Algorithm** | **AUC Average** | **Cohen's Kappa Coefficient** | **Matthews Correlation Coefficient** | **Loss** |
| LDA | 0.9889 | 0.7592 | 0.9072 | 0.0787 |
| D.T. | 0.9873 | 0.9384 | 0.9736 | 0.0229 |
| KNN | 0.9392 | 0.6864 | 0.8810 | 0.0961 |
| N.B. | 0.9935 | 0.8040 | 0.9225 | 0.0616 |
| SVM | 0.9988 | 0.9384 | 0.9757 | 0.0217 |
| Narrow-ANN | 0.9982 | 0.9552 | 0.9824 | 0.0136 |
| Medium-ANN | 0.9998 | 0.9552 | 0.9819 | 0.0147 |
| Wide-ANN | 0.9984 | 0.9440 | 0.9783 | 0.0165 |
| Bilayered-ANN | 0.9988 | 0.9552 | 0.9820 | 0.0147 |

Figure 6 shows the ROC curves of the top four ML algorithms trained for the classification of EEG signals related to the state of relaxation, right hand, left hand, both hands, and both feet. These algorithms are LDA, SVM, D.T., and N.B. The algorithm that presented the best performance metrics was SVM, with an AUC average of 0.9988. The ROC curves showed a compromise between sensitivity and specificity. The SVM algorithm was the closest to the upper-left corner of the ROC space, while the D.T. model was closer to the 45-degree diagonal. Classifiers that obtain curves closer to the upper-left corner indicate better performance, while classifiers with ROC curves closer to the 45-degree diagonal of the ROC space are less accurate.

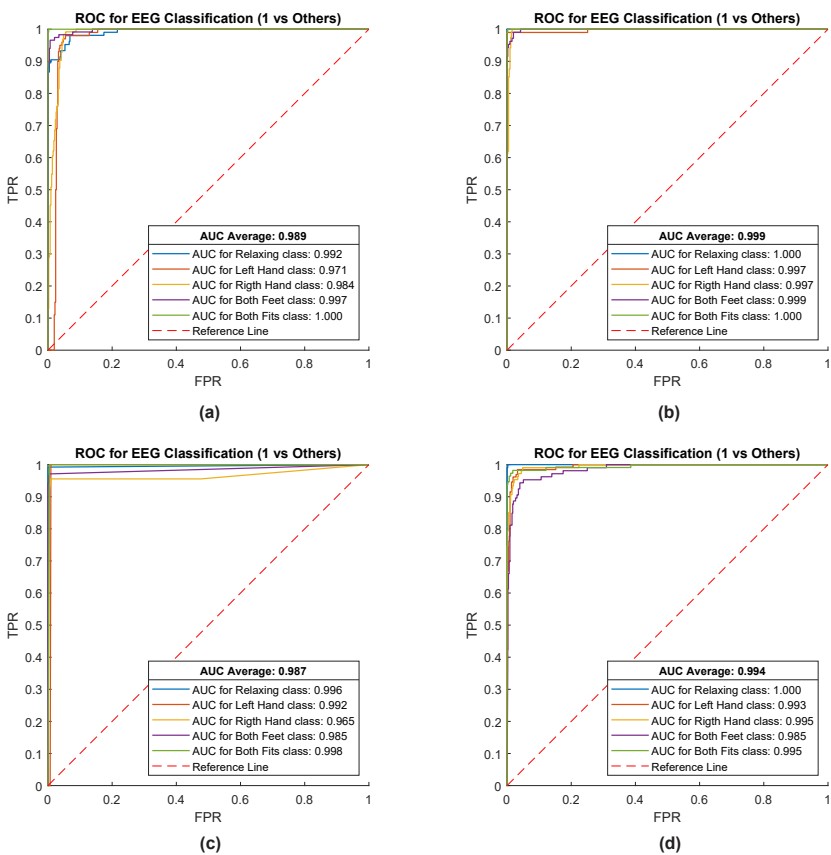

**Figure 6.** The receiver operating characteristic (ROC) curves of the top four ML algorithms trained for EEG signals classification, related to the movements of hands and feet: (**a**) ROC curves of the LDA algorithm, (**b**) ROC curves of the SVM algorithm, (**c**) ROC curves of the D.T. algorithm, and (**d**) ROC curves of the N.B. algorithm.

Figure 7 shows the ROC curves of the top four DL algorithms (neural networks) trained for the classification of EEG signals. These algorithms are Narrow-ANN, Medium-ANN, Wide-ANN, and Bilayered-ANN. The algorithm that presented the best performance metrics was Medium-ANN, with an AUC average of 0.9998; it was the closest to the upper-left corner of the ROC space.

In machine learning, the presumably best model is chosen from a collection of model candidates obtained by evaluating various model types, hyperparameters, or feature subsets, among others. In this paper, it is proposed to use ConfusionVis, a model-agnostic technique for evaluating and comparing multiclass classifiers based on their confusion matrices [56]. Figure 8 depicts the ConfusionVis achieved for the nine ML models chosen for EEG signal classification. Figure 8a shows the average accuracy score per ML model, where it can be observed that Narrow-ANN had the best accuracy score. Figure 8b illustrates the confusion matrix similarity results, where it can be seen that the D.T., SVM, Narrow-ANN,

and Medium-ANN models obtained the best similarity. Figure 8c depicts the error by class scores, where it can be observed that Medium-NN and Narrow-ANN achieved the lowest error score in most classes of movements classified from the EEG signals. Figure 8d shows the error by model scores, where it can also be seen that Medium-ANN obtained the lowest error score, followed by Bilayered-NN, Narrow-ANN, and decision tree (D.T.).

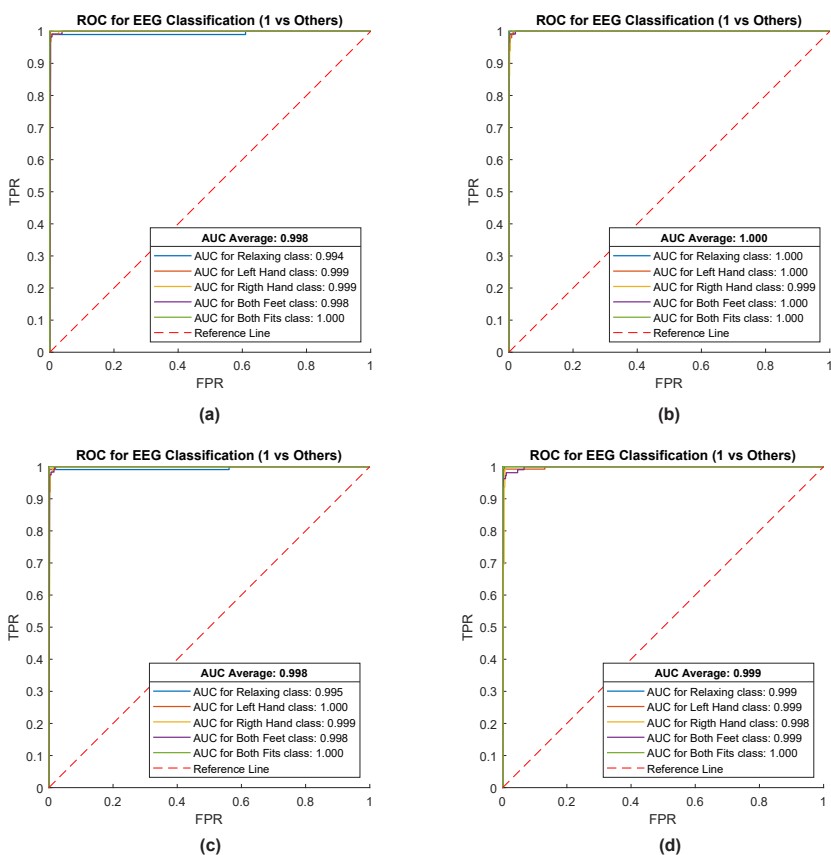

**Figure 7.** The receiver operating characteristic (ROC) curves of the four ANN algorithms trained for EEG signals classification, related to the movements of hands and feet: (**a**) ROC curves of the Narrow-ANN algorithm, (**b**) ROC curves of the Medium-ANN algorithm, (**c**) ROC curves of the Wide-ANN algorithm, and (**d**) ROC curves of the Bilayered-ANN algorithm.

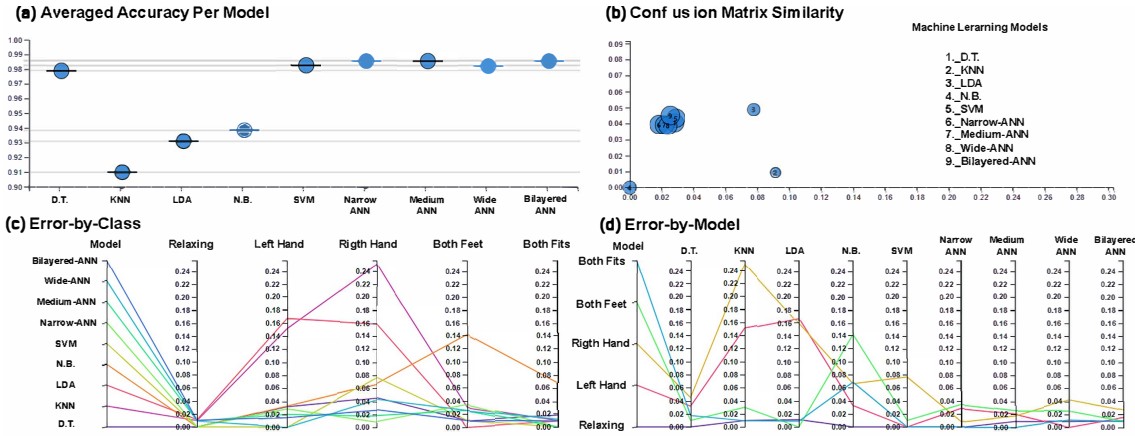

**Figure 8.** ConfusionVis [56]: Comparative evaluation of the multiclass classifiers based on confusion matrices. (**a**) Averaged accuracy per model, (**b**) confusion matrix similarity, (**c**) error by class, and (**d**) error by model.

Figure 9 shows the training time of the nine ML algorithms tested, with N.B., LDA, and KNN having the shortest training time. However, the results shown in Tables 4 and 5 show that these algorithms had the lowest performance metrics, with the exception of D.T. In contrast, the SVM, Narrow-ANN, Medium-ANN, Wide-ANN, and Bilayered-ANN algorithms had the most considerable training times of 0.13546, 0.37135, 0.16956, 0.36255, and 0.45722 s, respectively, with the Bilayered-ANN algorithm having the longest training time. However, these algorithms had the best performance metrics, as shown in Tables 4 and 5 and Figures 7 and 8. Therefore, the data science engineer or researcher must perform a cost–benefit analysis regarding accuracy and processing time. In most circumstances, engineers favor accuracy over training time, because training is only performed a few times and only the trained ML model is employed. For this reason, in this study, it is more convenient to select the Narrow-ANN model.

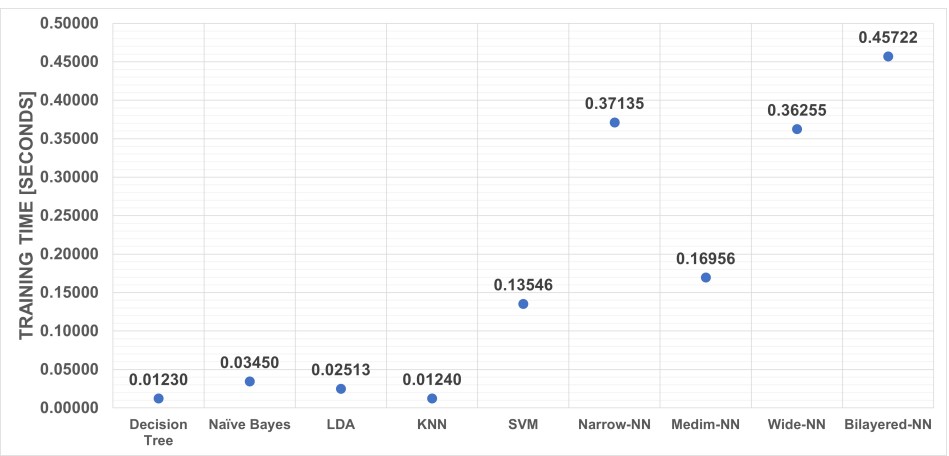

**Figure 9.** Training time of the nine ML models.

## 4. Discussion

In this study, we observed that the different features used were helpful for the classification of EEG signals, as proposed in our hypothesis. The presented features are based on the time domain: amplitude, frequency, phase, peak–peak value, negative peak, positive peak, median, mode, average, mean square error value, standard deviation, summation, variance, kurtosis, and skewness. We consider that they are good features for classifying EEG signals related to movements. Using these features, the ML model that achieved the best performance was Medium-ANN, with average area under the curve of 0.9998, Cohen's Kappa coefficient of 0.9552, Matthew correlation coefficient of 0.9819, and loss of 0.0147.

We observed that the performance metrics obtained from the nine machine learning algorithms were good. Using standard features in different frequency bands and related to a particular class allowed machine learning and deep learning algorithms to obtain excellent performance metrics, as shown in Tables 4 and 5 and Figures 6–8; this is because the proposed frequency bands and features improved the separability of the data, making the classification algorithms substantially better.

Regardless, the data science engineer/scientist is in charge of carrying out the corresponding analysis in terms of costs–benefits and precision concerning the information processing time. In most cases, ML models with better precision are chosen, and training time is usually sacrificed. Since the training of the ML algorithms is performed once, only the trained model is used for the assigned task. The Medium-ANN algorithm was selected for this reason and because its performance metrics were the best. Therefore, feature extraction is worth mentioning among the processes that improve relevant information acquisition and ensure better performance metrics when training EEG signal classification algorithms, as shown in different studies. Our results are consistent with other spectrogram methods implemented for identifying EEG patterns in persons with motor impairment using similar brain sources that were analyzed in this study [15]. Many human behavior

fields are still a challenge for BCIs; findings from this study may provide complementary data for other studies reporting findings from central nervous system damage with residuals of motor impairment of upper limb movement [18]. In addition to limb paralysis, limb loss represents an obstacle to quality of life for which the results of this study offer a comprehensive and reliable technique for extracting electrical brain sources for human movement programming. As in other research [20], results of the present study provide consistent and accurate information for future controlling inputs for the adaptation of prosthesis. As reported elsewhere [69], we conclude that it is necessary to increase movement classes in EEG features extraction for providing mechatronic systems controlled by means of BCI, suitable and reliable patterns corresponding for target movements.

## 5. Proposed Usage Scenario

The ML algorithms proposed in this research study could be implemented in high-performance embedded systems or edge computing devices as verified in previous studies [59,89]. These act as the central control system, which is in charge of communicating with the BCI to acquire EEG signals. Likewise, the control system is in charge of carrying out the digital processing of the EEG signals, the extraction of features, the classification, and the translation (decoding and execution) of the control commands. The mechatronic control system would have a trained ML model which would allow a user with some motor disability to perform some motor activities, such as opening and closing the right fist, left fist, or both fists through the classification of EEG signals.

Figure 10 depicts a conceptual diagram of the prospective mechatronic control system. We could consider this model the first step in developing intelligent prostheses that integrate the system's several components. The future characteristics to be developed are lower cost, size, portability, low power consumption, and reliable communication with the BCI.

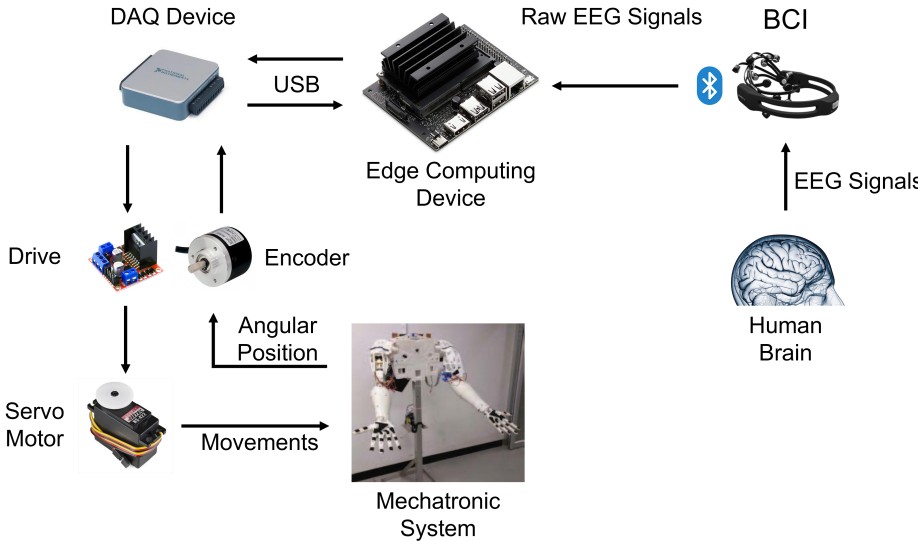

**Figure 10.** Suggested usage scenario for an application of mechatronic control system.

*Limitations of the Study*

One of the drawbacks of this research study is the need for a BCI; users should have short hair, as the BCI must be comfortable and enjoyable to them. Furthermore, the electrodes must be maintained in saline solution. Successful implementation also relies on the BCI battery life. Finally, if the emotional state of the participants is altered, accurate measurements cannot be acquired.

## 6. Conclusions

In this study, a methodology for classifying motor movements by processing the EEG signals of 30 users is presented. The classification of the EEG signals was related

to left hand, right hand, both fists, feet, and relaxation movements. As a result of EEG signal processing, a customized dataset was created and used to train the ML algorithms. The dataset was obtained by reading the EEG signal files in EDF+ format, extracting the different segments of the EEG signals, filtering the signals, extracting the features, and labeling their corresponding classes. The customized dataset was created to train and evaluate the performance metrics of different ML algorithms in the classification of EEG signals related to motor movements. The model of Medium-ANN achieved the best performance metrics, with an AUC average of 0.9998, Cohen's Kappa coefficient of 0.9552, Matthews correlation coefficient of 0.9819, and loss of 0.0147. These findings enable the approach to be applied to different scenarios, such as robotic prosthesis implementation, where the utilization of physical qualities is an acceptable alternative when hardware resources are restricted, or in embedded systems or edge computing devices, which have the advantages of low cost, small size, portability, low power consumption, and reliable communication with the BCI.

Furthermore, with the proposed method, we estimate that quantifiable information about motor movement can be obtained through the feature extraction and performance metrics of ML algorithms. We also believe that the proposed method could allow us to generate different datasets that could be used for future studies, as the proposed software was developed and customized to analyze EEG signals.

**Supplementary Materials:** The following supporting information can be downloaded at: https://www.mdpi.com/article/10.3390/technologies10040079/s1.

**Author Contributions:** Conceptualization, E.E.G.-G. and E.I.-G.; Data curation, J.M.C.-V., E.G.-C. and O.R.L.-B.; Formal analysis, E.E.G.-G., E.T.-C. and E.G.-C.; Investigation, F.J.R.-A., E.T.-C., G.M.G.-A. and E.I.-G.; Methodology, F.J.R.-A., G.M.G.-A. and E.I.-G.; Project administration, O.R.L.-B.; Resources, F.J.R.-A., O.R.L.-B. and E.I.-G.; Software, F.J.R.-A.; Supervision, E.E.G.-G. and E.I.-G.; Validation, J.M.C.-V., E.G.-C. and G.M.G.-A.; Visualization, J.M.C.-V. and O.R.L.-B.; Writing—original draft, F.J.R.-A.; Writing—review and editing, E.E.G.-G., E.T.-C., E.G.-C. and E.I.-G. All authors have read and agreed to the published version of the manuscript.

**Funding:** This research study's funds were provided by Universidad Autónoma de Baja California (UABC) through grant number 679.

**Institutional Review Board Statement:** Not applicable.

**Informed Consent Statement:** Not applicable.

**Data Availability Statement:** We share the dataset as supplementary material.

**Acknowledgments:** The authors are very grateful to PRODEP for supporting the academic groups to increase their degree of consolidation. We also want to thank the Faculty of Engineering and Technology Sciences (FCITEC) for all the facilities provided for the development of this project. F.J.R.-A. would like to thank SPSU-UABC (Union of Teachers University Improvement of the Autonomous University of Baja California) for the scholarship supporting their postgraduate studies.

**Conflicts of Interest:** The authors declare no conflict of interest. The funders had no role in the design of the study; in the collection, analyses, or interpretation of data; in the writing of the manuscript, or in the decision to publish the results.

## Appendix A. Fragment of the Dataset Created for This Study

| C3_Amplitude | C3_Frequency | C3_Phase | C1_Peak to Peak | C1_Negative Peak | C1_Positive Peak | Cz_Median | Cz_Mode | Cz_Mean | C2_RMS | C2_SD | C2_Summation | C4_Variance | C4_Kurtosis | C4_Skewness | Classes |
|---|---|---|---|---|---|---|---|---|---|---|---|---|---|---|---|
| 108.86617 | −29.32351 | 79.54266 | 0.26955 | 2.97236 | 1.51095 | 17.27037 | 17.22153 | 979.79113 | 295.73598 | 7.49178 | 1.60906 | 8.42745 | 0.01242 | −57.88825 | 0 |
| 145.2712 | −52.97385 | 92.29735 | −1.91859 | −11.14636 | −0.38611 | 18.46683 | 18.46945 | −543.63732 | 366.13252 | 5.52692 | 0.902 | 9.11907 | 0.00636 | 7.69257 | 0 |
| 100.73007 | −38.61309 | 62.11698 | −1.21691 | 0.21219 | −0.23162 | 16.3806 | 16.38057 | −951.47539 | 257.03075 | 3.80701 | 0.61077 | 8.92251 | 0.00794 | 24.9861 | 0 |
| 119.87798 | −57.16337 | 62.71462 | −1.69857 | 0.04493 | −0.00493 | 19.4224 | 19.42449 | 18.93081 | 368.71271 | 3.19239 | 0.32758 | 8.68137 | 0.0046 | −97.59919 | 0 |
| 125.2563 | −57.19535 | 68.06094 | −1.91499 | −5.52267 | −0.49701 | 19.50148 | 19.49685 | −2956.4001 | 377.8044 | 3.11134 | 0.27458 | 8.8037 | 0.00691 | −109.72379 | 0 |
| 125.67379 | −57.61492 | 68.05887 | −1.47821 | −4.39237 | 0.02402 | 19.54563 | 19.54696 | 173.02863 | 379.61249 | 3.12059 | 0.25601 | 7.58664 | 0.00088 | −37.67507 | 0 |
| 125.68931 | −57.61504 | 68.07427 | −1.74668 | −4.33893 | −0.02686 | 17.26488 | 17.26585 | −284.69096 | 298.30936 | 3.5752 | 0.41033 | 6.55339 | 0.00225 | 150.20889 | 0 |
| 125.31995 | −57.61353 | 67.70643 | −1.70869 | −4.66525 | 0.03645 | 16.42054 | 16.42135 | 213.89227 | 270.07081 | 3.84392 | 0.46781 | 5.07003 | 0.00898 | −129.32524 | 0 |
| 60.11105 | −26.70885 | 33.4022 | −0.69864 | −0.25313 | −0.5557 | 13.09898 | 13.09417 | −414.86556 | 162.42172 | 2.4878 | 0.25535 | 10.79111 | 0.01569 | 2.81889 | 0 |
| 190.91658 | −95.98303 | 94.93355 | −1.77143 | −5.3341 | −0.20248 | 20.87957 | 20.88366 | −434.88338 | 437.30133 | 8.8172 | 0.0715 | 16.00605 | 0.00621 | 115.23688 | 0 |
| 81.42884 | −35.41586 | 46.01297 | −1.01522 | −0.04502 | −0.15462 | 13.69927 | 13.69939 | −766.25766 | 193.38404 | 4.47048 | 0.48256 | 5.48752 | 0.01028 | −51.56381 | 0 |
| 149.91035 | −80.69967 | 69.21068 | −1.06843 | −9.09432 | −0.07855 | 20.50509 | 20.50722 | −237.70391 | 422.09271 | 4.16793 | 0.10647 | 8.36332 | 0.00551 | 140.77191 | 0 |
| 168.19124 | −80.77541 | 87.41583 | −1.01319 | 2.02131 | −0.17725 | 22.09124 | 22.09226 | −1142.61373 | 488.73057 | 4.06269 | 0.24355 | 9.24432 | 0.00596 | 10.96383 | 0 |
| 178.18538 | −80.95155 | 97.23383 | −1.91043 | −1.48237 | −0.38365 | 23.20017 | 23.19817 | −2985.49963 | 542.32634 | 4.07673 | 0.37249 | 6.94932 | 0.007 | 124.37271 | 0 |
| 165.99797 | −68.76385 | 97.23412 | −1.80815 | 3.15668 | −0.28765 | 24.48947 | 24.48931 | −2400.00755 | 603.6636 | 4.07886 | 0.54094 | 8.45258 | 0.00216 | 21.81982 | 0 |
| 164.91541 | −67.67608 | 97.23933 | −1.7938 | 2.43564 | 0.06713 | 24.04032 | 24.04147 | 510.43837 | 585.46381 | 4.08828 | 0.56694 | 10.81424 | 0.00215 | −45.78993 | 0 |
| 208.80963 | -113.29443 | 95.5152 | 0.15291 | −17.76571 | 3.27554 | 40.61923 | 40.51458 | 2222.06571 | 1621.20376 | 3.29414 | −0.15438 | 39.40843 | 0.0081 | −115.65555 | 0 |
| 86.64714 | −32.25059 | 54.39655 | −1.54822 | −1.8633 | −0.34174 | 12.82435 | 12.8229 | −690.91512 | 159.61514 | 4.44967 | 0.70982 | 7.62719 | 0.00757 | 200.835 | 0 |
| 85.72714 | −36.61694 | 49.1102 | −1.34702 | 0.03854 | −0.03464 | 14.51275 | 14.51493 | 37.44572 | 206.20682 | 3.25947 | 0.48182 | 7.04131 | 0.00579 | 64.3528 | 0 |
| 96.2026 | −48.3727 | 47.8299 | −1.55971 | −0.02817 | −0.13503 | 14.57802 | 14.57878 | −715.35163 | 207.82778 | 3.31905 | 0.48414 | 7.16151 | 0.00606 | −119.52097 | 0 |
| 96.61122 | −48.34653 | 48.26469 | −2.07812 | −3.30589 | −0.16501 | 15.34706 | 15.3469 | −1257.20859 | 232.60851 | 3.21864 | 0.47966 | 7.57887 | 0.00601 | −81.37703 | 0 |
| 115.44138 | −48.12757 | 67.31381 | −1.68488 | −4.53478 | −0.09991 | 15.84409 | 15.84484 | −758.99683 | 250.80891 | 3.39552 | 0.49671 | 8.80287 | 0.00606 | −125.87036 | 0 |
| 107.7065 | −40.38883 | 67.31767 | −1.99927 | −3.99952 | −0.03327 | 16.32575 | 16.32665 | −337.47932 | 264.43673 | 3.53177 | 0.60548 | 9.58213 | 0.00605 | −106.34569 | 0 |
| 107.3927 | −40.07353 | 67.31917 | −1.95398 | −5.44053 | −0.08078 | 15.80524 | 15.8058 | −857.09752 | 246.15637 | 3.57552 | 0.57687 | 7.56182 | 0.00606 | −111.23568 | 0 |
| 129.15821 | −78.83456 | 50.32365 | 2.54778 | −1.75504 | 1.9225 | 25.30673 | 25.26614 | 1164.45591 | 642.98665 | 4.72533 | −0.92255 | 10.21739 | 0.00855 | −179.19851 | 0 |
| 105.01136 | −43.05677 | 61.95459 | −1.74764 | −2.98789 | −0.22264 | 18.15878 | 18.1624 | −366.03301 | 313.94184 | 3.64993 | 0.43932 | 11.21388 | 0.00608 | −49.82756 | 0 |
| 103.4699 | −39.51072 | 63.95918 | −0.33653 | 3.66172 | −0.0269 | 16.65289 | 16.65536 | −105.03978 | 262.71902 | 3.74805 | 0.49868 | 9.27736 | 0.00584 | −9.21414 | 0 |
| 170.39639 | −95.58504 | 74.81135 | −1.27524 | −2.79184 | −0.1146 | 18.94103 | 18.94256 | −642.32131 | 344.03532 | 4.73865 | 0.20353 | 8.07856 | 0.00629 | 43.58595 | 0 |
| 170.18116 | −95.78412 | 74.39704 | −1.66529 | −3.58502 | −0.05165 | 18.01691 | 18.01833 | −322.69376 | 315.32922 | 5.12737 | 0.08568 | 8.99191 | 0.00726 | −17.34837 | 0 |
| 132.33885 | −66.59612 | 65.74274 | −1.70012 | −4.83305 | −0.01402 | 16.44777 | 16.44886 | −260.7329 | 272.77158 | 3.90503 | 0.30343 | 6.0148 | 0.00643 | 233.40587 | 0 |
| 140.09181 | −66.24984 | 73.84196 | −1.52839 | −5.4838 | 0.03415 | 16.85112 | 16.85209 | 129.02571 | 289.59865 | 3.76946 | 0.35227 | 5.85945 | 0.00643 | 76.20641 | 0 |
| 129.81297 | −55.97095 | 73.84203 | −1.68358 | −2.65769 | −0.06647 | 16.31524 | 16.31593 | −642.59732 | 267.36145 | 3.97121 | 0.52604 | 6.43784 | 0.0083 | −185.97406 | 0 |
| 74.47058 | −33.17593 | 41.29465 | −1.06796 | −5.44768 | −0.00179 | 15.44494 | 15.45545 | −117.89723 | 219.18465 | 3.20773 | 0.39781 | 9.42659 | 0.01214 | 206.93585 | 0 |
| 101.9468 | −44.0935 | 57.8533 | −1.57687 | −3.03288 | −0.19162 | 16.61435 | 16.61664 | −497.86008 | 271.47804 | 4.08116 | 0.52031 | 7.4292 | 0.00666 | −29.1921 | 0 |
| 161.79353 | −97.85084 | 63.94269 | −1.63274 | −3.40556 | −0.08934 | 20.86456 | 20.86752 | −279.22265 | 416.11531 | 4.82303 | −0.00011 | 14.74093 | 0.00469 | 156.06326 | 0 |
| 168.28902 | −98.98957 | 69.29944 | −2.26611 | −4.3866 | −0.0091 | 18.43858 | 18.44054 | −121.97746 | 333.03526 | 5.78385 | 0.23606 | 8.07245 | 0.00565 | 34.34179 | 0 |
| 118.41162 | −49.11881 | 69.29281 | −1.80372 | 1.16639 | −0.16257 | 15.4754 | 15.47586 | −957.41073 | 238.87176 | 4.28581 | 0.82451 | 7.86934 | 0.00609 | −169.0244 | 0 |

**Figure A1.** Fragment of the dataset created for this study.

**Appendix B. Front Panel of Software (App) Developed for EEG Signal Analysis**

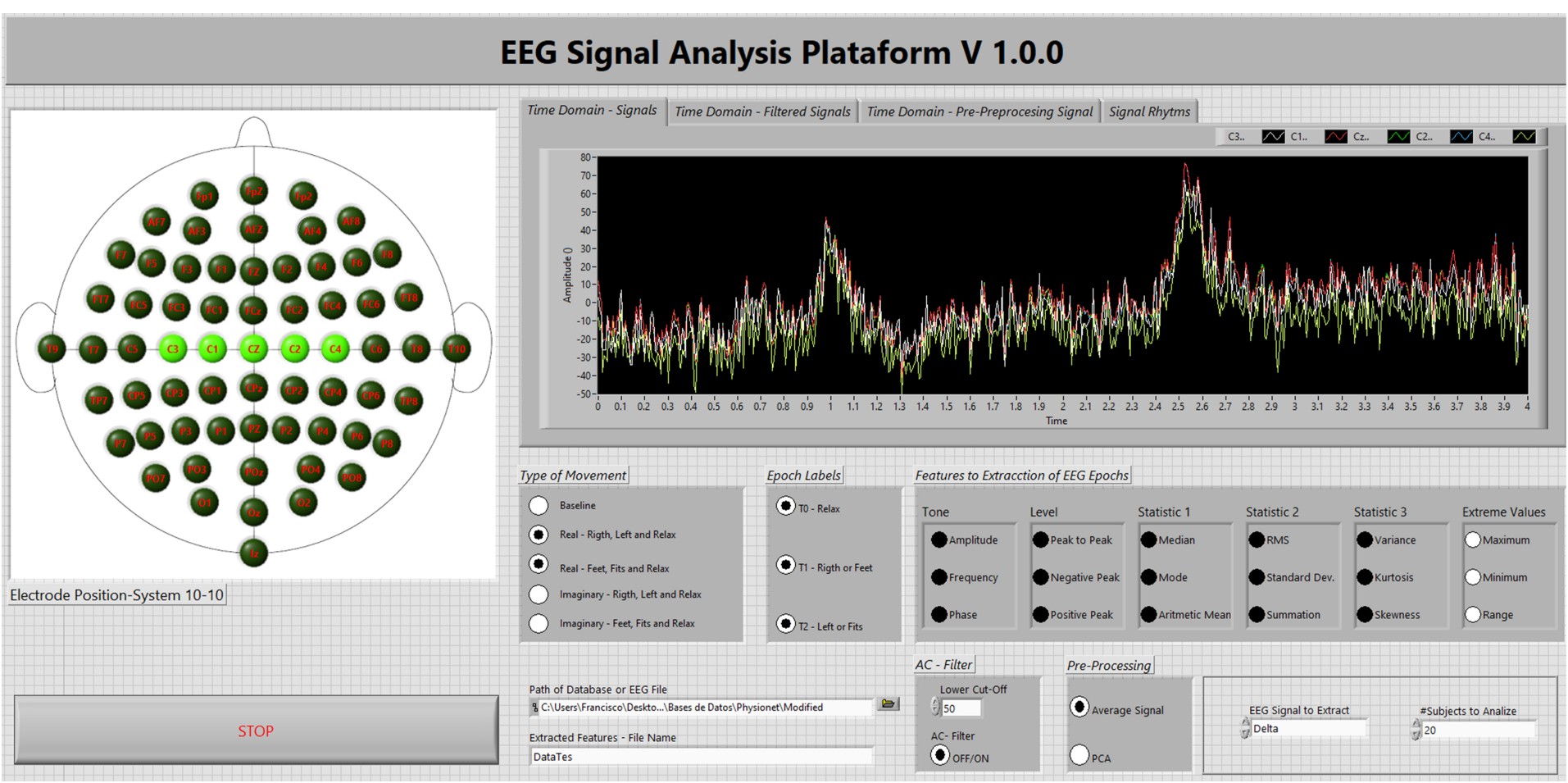

**Figure A2.** Front panel of software (App) developed for EEG signal analysis.

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
