# Peer review of "Evaluation of Machine Learning Algorithms for Classification of EEG Signals"

_technologies, doi:10.3390/technologies10040079_

Round 1

Reviewer 1 Report

1: This paper mainly focuses on traditional machine learning methods. The authors should add some deep learning methods, which are mentioned in the paper Toward open-world electroencephalogram decoding via deep learning: a comprehensive survey.

2: The quality of the figures in this paper should be improved.

3: Some deep discussions should be added in this paper.

4: There are some grammatical problems in the paper, please check and edit.

Reviewer 2 Report

This paper used some publicly available EEG data and applied machine learning to classify motor movement. The paper was written terribly, and, in my opinion, there is no significant contribution of this paper that could make this paper publishable. My comments are given below.

1.       The writing quality of the paper is terrible, there is no flow in writing, which makes it very annoying to read through. After reading the abstract I was not sure what the authors are trying to do. The abstract is rather unorganized and chaotic.

2.       The features used in the paper are very common, so I don’t see any innovations on that. Moreover, just using a bunch of ML methods doesn’t necessarily increase the novelty. I would assume there are plenty of papers already available on that. In that case comparison with existing papers should be provided to claim the superiority of their model.

3.       Even though the original dataset contains 109 subjects the authors finally used only 25 with only limited movement (binary?) which is rather small. In that doing a leave one subject out validation will be more preferable.

4.       It’s still confusing how many classes they have; it appears they have multiple movements; in that case how did they manage to calculate F1 score?

5.       The paper did not use proper references from lots of information they used in the introductions.

Round 2

Reviewer 1 Report

I suggest accepting this paper.

Reviewer 2 Report

The authors revised paper and improved the writing. The paper can be considered for publication for now.